# Impact of Vitamin D Deficiency on Tumor Aggressiveness in Neuroendocrine Neoplasms

**DOI:** 10.3390/nu15173771

**Published:** 2023-08-29

**Authors:** Manuela Albertelli, Cristian Petolicchio, Sara Brasili, Andrea Pogna, Mara Boschetti, Giorgio Luciano, Davide Campana, Stefano Gay, Alessandro Veresani, Diego Ferone, Lara Vera

**Affiliations:** 1Endocrinology Unit, Department of Internal Medicine and Medical Specialties (DiMI), University of Genova, 16132 Genoa, Italy; manuela.albertelli@unige.it (M.A.); dottorcristianpetolicchio@gmail.com (C.P.); sara.brasili.5a@gmail.com (S.B.); mara.boschetti@unige.it (M.B.); veresani.alessandro@gmail.com (A.V.); 2Endocrinology Unit, IRCCS Ospedale Policlinico San Martino, 16132 Genova, Italy; stefano.gay@hsanmartino.it (S.G.); lara.vera@hsanmartino.it (L.V.); 3CNR SCITEC “Giulio Natta” Scitec Istituto per Studio delle Science e Tecnologie Chimiche, 16149 Genova, Italy; giorgio.luciano@scitec.cnr.it; 4Bologna ENETS Center of Excellence, S. Orsola-Malpighi University Hospital, Alma Mater Studiorum, University of Bologna, 40126 Bologna, Italy; davide.campana@unibo.it

**Keywords:** vitamin D, vitamin D deficiency, 25(OH)D, neuroendocrine neoplasm, neuroendocrine tumors, aggressiveness, prognosis

## Abstract

**Background:** The role of vitamin D (25(OH)D) in the pathogenesis and outcome of several conditions, including autoimmune diseases, diabetes and cancers is largely described in the literature. The aims of this study were to evaluate the prevalence of 25(OH)D deficit in a cohort of patients with neuroendocrine neoplasms (NENs) in comparison to a matched healthy control group and to analyze the possible role of 25(OH)D as a prognostic factor for NENs in terms of biological aggressiveness, tumor progression and survival. **Methods:** From 2009 to 2023, 172 patients with NENs (99 females; median age, 63 years) were included in the study. Serum 25(OH)D levels were defined as deficient if ≤20 ng/mL. The possible associations between 25(OH)D levels and disease grading, staging, ki67%, overall survival (OS), and progression-free survival (PFS) were considered. **Results:** NEN patients had significantly lower 25(OH)D levels compared to controls (*p* < 0.001) regardless of the primary origin. Patients with 25(OH)D < 20 ng/mL had a significantly higher ki67 index (*p* = 0.02) compared to the ones with 25(OH)D levels above 20 ng/mL. Patients with disease progression were found to have a significantly lower 25(OH)D at baseline (*p* = 0.02), whereas PFS and OS were not significantly influenced by 25(OH)D. **Conclusions:** Vitamin D deficiency is highly prevalent among NENs and is associated with higher ki67 and disease progression. Our study highlights the importance of monitoring 25(OH)D levels in patients with NENs, as its deficiency appeared to be linked to the worst biological tumor aggressiveness.

## 1. Introduction

Neuroendocrine neoplasms (NENs) are a heterogeneous group of relatively rare tumors originating from neuroendocrine cells that can occur anywhere in the body, more frequently in the gastrointestinal tract and bronchopulmonary systems [1].

Since Siegfried Oberndorfer introduced the term carcinoid in 1907 [2] and the endocrine nature of these tumors was identified in 1914 [3], a lot of scientific advances [4], improved disease classification [5,6] and better understandings of those tumors were made.

NENs have a heterogeneous spectrum, ranging from indolent well-differentiated neuroendocrine tumors (NETs) to very aggressive poorly differentiated neuroendocrine carcinomas (NECs) [7,8]. Despite the diversity in tissue origin, all these tumors typically share common morphological and immunophenotypical features, including cellular function, growth pattern and expression of neuroendocrine markers.

Although they may arise in the setting of hereditary syndromes, such as the multiple endocrine neoplasia type 1 and 2A (MEN-1 and MEN-2A) [9,10], von Hippel-Lindau disease [11], tuberous sclerosis type 2 [12] and neurofibromatosis type 1 [13], most of the NENs are sporadic and NENs risk factors have not yet been fully elucidated [14].

Actually, there are many effective therapy strategies for NEN: in a randomized clinical trial (RCT) of neuroendocrine tumors, 22 different therapeutic strategies were compared, each with a different safety profile and different impact on 25(OH)D levels [15].

The role of vitamin D in the pathogenesis and outcome of several conditions, including autoimmune diseases, cardiovascular disease, diabetes and cancers is largely described in the literature [16,17,18,19,20,21]. 

In in vitro studies, vitamin D and its analogues such as 25-hydroxyvitamin D (25(OH)D) have been shown to be involved in cell growth, apoptosis, cell signaling, differentiation, cell adhesion, cell metabolism, immune regulation, angiogenesis, and metastasis [22,23,24].

Also, some in vivo studies have been carried out on the vitamin D impact especially on highly immunological tumors, such as melanomas and prostate cancer [25,26].

Current evidence suggests that 1,25OH-vitamin D3 and its analogs can inhibit tumor cells growth to its anti-proliferative and pro-differentiating effects, as well as by influencing inflammation [27].

In addition, it has been recently demonstrated that circulating levels of 25(OH)D above 21.6 ng/mL and 30 ng/mL may contribute to reducing cancer mortality and cancer risk, respectively [28,29].

Many factors impact vitamin D levels, among them are environmental conditions such as climate temperate and geographical latitude or medical issues such as metabolic disorders, diabetes and obesity. Recent evidence highlights that multiple deficiencies of bioactive compounds, antioxidants and flavonoids might have a role in reducing cancer risk [30].

In recent years, some studies have evaluated vitamin D levels in NENs and specifically in gastroenteropancreatic NENs (GEP-NENs), most of them demonstrating a significantly higher prevalence of vitamin D deficit in those patients [31,32,33,34,35,36].

Some studies confirm that a vitamin deficit has a high prevalence in GEP-NEN; in fact as a result of the illness, the consumption of some nutrients can be reduced, leading to malnutrition or nutritional deficiencies such as vitamin E, folates and niacin [30].

An interesting review analyzed evidence on the mechanisms that could have a potential impact on vitamin D levels in GEP-NETs such as hormone hypersecretion, specific microRNAs and nutritional status, which can be affected by the neoplasia, the treatment with somatostatin analogues and the surgical tumor resection [35]. Previously, despite the small number of subjects analyzed, another study demonstrated that primary tumor localization does not influence vitamin D levels in patients with NET [36]. Therefore, the aims of this study are to evaluate the prevalence of 25(OH)D deficit in a well-characterized cohort of NEN patients in comparison to a matched healthy control group and to analyze the possible role of 25(OH)D values as a prognostic factor for NENs in terms of biological aggressiveness, tumor progression (PD) and survival, regardless of primary tumor localization.

## 2. Subjects and Methods

### 2.1. Study Design and Population

We conducted an observational retrospective cohort study on a population of 172 out of 414 patients affected by NEN followed up at the Endocrinology Unit of the IRCCS Ospedale Policlinico San Martino, University of Genova, over a time range of 14 years, from 2009 to 2023.

To improve the power of the study, we included only patients with the following criteria:-Presence of 25(OH)D value at diagnosis;-Age > 18 years old;-Absence of primary hyperparathyroidism (PTH levels above the upper limit due to dysregulated production by one or more hyperplastic parathyroids);-Absence of chronic renal or liver insufficiency and inflammatory diseases;-BMI < 35 kg/m^2^;-Well differentiation at cytological/histologic examination;-Absence of chronic use of medicaments or supplements known to interfere with 25(OH)D metabolism (including cinacalcet, calcium, bone antiresorptive therapies, sex-hormones and anti-inflammatories).

The flow chart of the studied subjects is shown in Figure 1.

The clinicopathological characteristics considered were sex, age at diagnosis, body mass index (BMI), primary neoplasm site, grading, stage, proliferation index with value of ki67% and functional status of tumors. The histological characteristics of the neoplasms were obtained after surgical resection or biopsy. Tumor grade was classified according to the World Health Organization (WHO) classification as G1 (ki67% ≤ 2% and/or <2 mitoses/mm^2^), G2 (ki67% 3–20% and/or 2–20 mitoses/mm^2^) or G3 (ki67% > 20% and/or >20 mitoses/mm^2^) [6].

The biochemical parameters considered were 25(OH)D at the time of diagnosis, calcium, phosphorus, parathormone (PTH), albumin and creatinine. 

The levels of 25(OH)D was evaluated both as absolute value and as relative value in line with the Endocrine Society guidelines and the report of the British Scientific Advisory Committee on Nutrition: 25(OH)D severe deficiency was defined as serum concentration of 25(OH)D < 12 ng/mL, 25(OH)D deficiency < 20 ng/mL, insufficiency as levels between 20 and 30 ng/mL and normal levels for values ≥ 30 ng/mL [37,38].

Data regarding tumor history as far as follow-up duration, time of progression-free survival (PFS), time of overall survival (OS) and vitamin D supplementations were also collected.

The control group was composed of healthy volunteers with a random 25(OH)D value. None of the volunteers enrolled had a history of neoplasia, inflammatory diseases, liver or renal failure or primary hyperparathyroidism. To improve the power of the study, the group was matched for age, sex, BMI, and geographical provenience with the patients of the NEN group and was finally formed by 50 volunteers.

The study was approved by the local Ethical Committee and was conducted in accordance with the principles of the Declaration of Helsinki. Informed consent was obtained from all subjects involved in the study.

### 2.2. Laboratory Assessment

In all patients (NEN and controls), the 25(OH)D and PTH levels were measured using chemiluminescence immunoassay (LIASON, DIA, Sorin kit). The coefficient of variation (CV) intra-dosage and inter-dosages was 6% for both assays. Normal PTH range was considered to be between 14 and 65 pg/mL at our laboratory unit.

Calcium, phosphorus and creatinine were measured through colorimetric standard techniques on heparinized blood samples.

### 2.3. Statistical Analysis

The descriptive analysis was performed by using frequency, mean, range, median, interquartile range (IQR 25°–75°) and standard error.

To assess the differences between groups, we conducted both parametric and non-parametric tests. For parametric data with normal distribution, we employed the independent samples *t*-test for two-group comparison and one-way analysis of variance (ANOVA) for multiple group comparisons.

Conversely, non-parametric data were analyzed using the Mann–Whitney U test for two-group comparisons and the Kruskal–Wallis test for multiple group comparisons.

To examine the association between categorical variables, we used Pearson’s chi-squared test (χ^2^).

PFS was defined as the interval between the diagnosis of NEN and the disease progression. The OS was defined as the interval between the diagnosis and them being deceased. To evaluate the survival outcomes of patients and compare survival distributions among different groups using log-rank tests, we employed Kaplan–Meier survival curves.

The risk factors, which could predict the disease progression, were evaluated through univariate and multivariate analysis using Cox’s proportional risks method. Risk factors were expressed as Hazard Ratio (HR) and a 95% Confidence Interval (CI) was used. The multivariate analysis was performed by using the forward stepwise method, after having excluded all the variables.

We employed various statistical methods using the R studio programming language to analyze clinical data (URL: https://www.rstudio.com/, accessed on 13 July 2023) and a dedicated software IBM—SPSS Statistics v. 22 [39,40,41].

A *p* value < 0.05 was considered statistically significant using both parametric and non-parametric tests.

## 3. Results

### 3.1. Clinical and Pathological Features

A population of 172 patients with a diagnosis of NEN was retrospectively evaluated and compared with a control group composed of 50 healthy volunteers, matched for age, BMI and geographical provenience.

In the NEN group, 58% of patients were women and 42% were men, with a male–female ratio of 1:1.4. The mean age at diagnosis was 59.7 years old and the mean BMI was 24.3 ± 0.3 kg/m^2^. 

In the control group, 57% of patients were women and 43% were men. The mean age at diagnosis was 61.2 years old and the mean BMI was 23.4 kg/m^2^.

Our cohort was formed of 72.1% patients with GEP-NEN, 19.2% with thoracic NEN and 15 patients (8.7%) where the localization of the primary tumor remained unknown. The most common NEN site was the pancreas (*n* = 58), followed by the ileum (*n* = 41), lungs (*n* = 30), stomach (*n* = 12), appendix (*n* = 8), colon (*n* = 5) and thymus (*n* = 3). Most patients had non-functioning NEN, whilst 50 patients (29.1%) clinically presented the associated-neuroendocrine syndrome. Metastasis was found in 44.2% (*n* = 76) of patients at diagnosis, whilst 37.3% (*n* = 64) had more than one lesion.

Among patients affected by GEP-NENs who underwent surgery, 69 (61%) had a neoplasm classified as grade 1 (G1), 42 (37%) as grade 2 (G2) and 2 (2%) as grade 3 (G3). Thirty patients had a lung NEN (22 typical, 73%, and 8 atypical carcinoids, 27%) and 3 patients had a thymic NEN (all atypical carcinoids). Data regarding the tumor grading were not available for 26 patients.

At the time of diagnosis, 23% (*n* = 39) were treated with vitamin D supplementation (cholecalciferol 1000–2000 unit daily).

The clinical and pathological features of NEN are shown in Table 1 (discrete variables) and in Table 2 (continuous variables).

### 3.2. Levels of 25(OH)D

As far as the biochemical characteristics, at the time of diagnosis, the median levels of calcium were 9.5 ± 0.04 mg/dL, of PTH were 33 ± 2.7 pg/mL, of phosphorus were 3.1 ± 0.06 mg/dL and of creatinine were 0.9 ± 0.05 mg/dL, all within the normality range.

Date regarding 25(OH)D levels are shown in Table 1 and Table 2.

In all patients we evaluated supplementation therapy with cholecalciferol 1000–2000 unit daily: 25(OH)D levels were significantly higher in patients assuming supplementation than in patients not assuming supplementation (median 26.04 ± 1.55 ng/mL vs. 14.54 ± 0.73 ng/mL, *p* < 0.001). In patients treated with vitamin D supplementation, 18% (*n* = 7) have not reached adequate 25(OH) levels at our evaluation (25(OH)D < 20 ng/mL). 

Regarding 25(OH)D levels, at baseline the NEN group had significant lower median levels of 25(OH)D (15.0 ± 0.75 ng/mL), compared to the control group, 30.5 ± 2.05 ng/mL (*p* < 0.001). Moreover, a higher prevalence of 25(OH)D deficit in patients with a diagnosis of NEN (87.2%) was evident, compared to the control population (21.1%; *p* < 0.0001). Particularly, 25(OH)D deficiency and severe deficiency was observed in 62.2% and 32% of NEN, compared to 14.8% and 6.3% in controls, respectively (χ^2^ = 35.9, *p* < 0.0001); (Figure 2 and Figure 3). Only 12% of NENs had vitamin D sufficiency compared to 48.9% of control cases (Figure 3). In addition, this difference was not result significantly influenced by the localization of primary NEN (*p* = 0.56). In fact, patients with GEP-NEN and NEN of unknown primary tumor localization were found having lower 25(OH)D levels (median = 16.5 ng/mL for both) compared to the thoracic ones (median = 19.4 ng/mL) but without a statically significance (*p* = 0.42), as shown in Figure 4. Even stratifying the patients according to the site of primary origin (pancreas, ileum, stomach, appendix, colon, lung, and thymus), no statistically significant differences emerged for the levels of 25(OH)D (*p* = 0.56).

### 3.3. Tumor Aggressiveness and 25(OH)D Levels

According to Endocrine Society guidelines and the report of the British Scientific Advisory Committee on Nutrition, we consider the 25(OH)D cut-off values of 12 ng/mL (severe deficiency) and 20 ng/mL (deficiency) at the time of diagnosis, and we therefore evaluated biological aggressiveness [37,38].

Mean ki67 of the entire cohort was 5.1% ± 0.9 (Table 2). Median ki67 of patients with 25(OH)D deficiency < 20 was 5.4% whilst median ki67 for patients with 25(OH)D > 20 ng/mL was 2.92% and none of them had ki67 higher than 10%. Patients with 25(OH)D < 20 ng/mL had significantly higher ki67 (*p* = 0.02) compared to the ones with 25(OH)D levels above 20 ng/mL, as shown in Figure 5. 

Instead, no significant correlation was found between 25(OH)D deficit and the number of metastases, the staging or the grading of the disease.

Regarding grading, mean 25(OH)D levels did not differ between patients with G2 tumors compared to those with G1 (17.0 ± 9.5 vs. 16.1 ± 10.2 ng/mL). Particularly, 44.2% and 67.4% of G2 patients had a vitamin D severe deficiency or deficiency, compared to 35.2% and 64.7% of G1 patients, respectively (χ^2^ = 0,98 *p* = 0.32, and χ^2^ = 0, *p* = 0.96, respectively for severe deficiency and deficiency).

### 3.4. Clinical Outcomes and 25(OH)D Levels

Regarding the neoplastic history, patients were evaluated for a median follow-up period of 37.5 months (1–372 months). In this period, PD was documented in 52 (30.8%) patients and 18 (10.5%) of them died as a consequence of it. Patients with PD were found to have a significantly lower 25(OH)D at baseline (*p* = 0.02), as shown in Figure 6.

As far as the deceased were concerned, considering the low number of events, no significant difference was found depending on 25(OH)D levels. 

PD and exitus were not significantly influenced by vitamin D supplementation. 

In our cohort, neither the presence of 25(OH)D deficiency (*p* = 0.52) nor its supplementation (*p* = 0.09) resulted in influencing the PFS. Even the overall survival (OS) was not result significantly influenced by 25(OH)D levels or supplementations.

In the univariate analysis, low 25(OH)D levels were found to be a risk factor for disease progression, together with the ki67 index and the stage of the disease at the diagnosis. Despite this finding, 25(OH)D levels did not confirm its role as a predictor of progression in the multivariate analysis, as no statistical significance was reached. Only ki67 and staging of the disease were confirmed as significant predictors in the multivariate analysis (Table 3).

## 4. Discussion

In this retrospective study, a heterogeneous cohort of NEN patients with different primary tumor localization and a medium–long-term follow-up was analyzed and compared to a healthy control group.

We confirmed that 25(OH)D deficiency is significantly more frequent among NENs patients than in the healthy population, independently from the localization of a primary tumor. Deficiency of 25(OH)D is a well-known condition for many neoplastic patients, common to different types of solid tumors and often linked to the nutritional status and to behavioral habits [42,43]. On the other hand, the 25(OH)D deficit may also appear as a possible risk factor for the development of neuroendocrine tumors. Many epidemiological studies, in fact, have already shown an inverse association between 25(OH)D levels and breast, colon, prostate and lung cancers [23,44,45]. Also, a recent metanalysis published by Arayici et al. confirmed higher serum 25(OH)D levels being associated with a lower general cancer risk [46]. The analysis results were especially strong for colorectal cancer, as the data were also confirmed by two other meta-analyses conducted by Hernandez-Alonso et al. and Lopez-Caleya et al. [47,48].

Our cohort was formed of both GEP-NEN and thoracic NEN, though in 15 patients, the localization of the primary tumor was not identified. 

In the whole NEN group the prevalence of 25(OH)D deficits was 86.7% and 61.7% had 25(OH)D deficiency (defined as 25(OH)D ≤ 20 ng/mL). These data are in line with evidence from the literature, where 25(OH)D deficiency in patients with NEN is described between 46% and 81% [43].

Even when stratifying patients and controls into different category of 25(OH)D deficit according to deficit severity, NEN patients had a more severe deficit than controls (32% of NEN patients vs. 6.3% in controls) (Figure 1 and Figure 2).

It is notable that no significant differences in 25(OH)D levels were found between the different categories of NEN, as GEP-NEN and NEN of unknown primary origin had lower median levels compared to thoracic NENs, but without statistical significance. The higher prevalence of 25(OH)D deficit has been previously demonstrated for GEP-NETs [32,34], where surgical treatment, steatorrhea and poor nutritional status are all linked to the malabsorption causing lower levels of nutrients and fat-soluble vitamins [43]. NENs of unknown primary tumor site can also be assimilated to GEP-NEN, as they are often located in the small bowel, especially in the ileum tract, which is the most difficult to investigate with current diagnostic methods. Only rare studies, such as the one conducted by Motylewska et al., have so far considered GEP-NEN together with NEN with different primary tumor localizations, showing no differences in 25(OH)D levels among the various NEN histotypes, although in a small sample of cases [36]. Even in our cohort, these results were confirmed.

Another interesting finding is that 25(OH)D deficiency appeared to be associated with the biological aggressiveness of the tumor, expressed as the ki67 value. A correlation between the ki67 index and 25(OH)D deficiency so far has been analyzed with regard to other types of solid tumor, supporting this possible correlation [49,50]. For instance, Wagner et al. demonstrated that the calcitriol level attained in patients with prostatic neoplasia was inversely associated with ki67 labeling [51].

Despite this finding, and differently from Altieri et al., which previously evidenced that 25(OH)D deficiency was associated with a higher grade tumor at the diagnosis [34], in our cohort, no significant correlation was found between grading and 25(OH)D levels. 

This could be due to the broad spectrum of ki67 values included in the grading categories, which account for the difference between the results of the ki67 evaluated as a continuous variable and the ki67 considered as a categorical variable. In fact, in our cohort the mean ki67 value of the 25(OH)D deficiency group is 5.37%, which is classified as G2 according to WHO classification; the mean ki67 value of 25(OH)D not-deficiency group is 2.92%, which is nearly identical to the lower limit of the G2 grading category. 

In addition, for other localizations of NEN, grading is not only based on ki67 levels, but also on the number of mitoses like thoracic NENs [6,52].

We also take into account the low number of G3 patients in our cohort, as suggested by Massironi at al. [32]. In all the studies mentioned above, 25(OH)D deficiency was not related to tumor staging, as also confirmed by our data [32,34]. This last parameter, in fact, is more influenced by the delay in the diagnosis rather than the intrinsic characteristics of the tumor.

Biological tumor aggressiveness seems therefore to be associated with 25(OH)D levels. A possible explanation might be suggested by looking at the pleiotropic effects of vitamin D as a potential role player in the evolution of the aggressiveness of the tumor, by acting on molecular signaling, neoangiogenesis and immunological responses. For example, vitamin D has been found to be able to act over the expression of cell cycle inhibitors p21 and p27 [53] and to regulate the activity of the cell adhesion molecules E-cadherin and β-catenin [54,55,56]. Also, it has been demonstrated that the vitamin D receptor (VDR) is expressed by various cell types, including bone marrow, breast, colon and immune cells (such as monocytes, macrophages and dendritic cells) [57,58]. Several studies confirmed the correlation between 25(OH)D deficiency and reduced immune function or autoimmunity development [59,60,61]. For example, in vivo studies showed that in VDR knock-out mice, the invariant Natural Killer T (iNKT) cells are functionally defective and secrete significantly less cytokines, since they fail to develop to mature iNKT [62]. Also, the expansion and proliferation of early iNKT cell precursors showed to be regulated by both vitamin D and the VDR [62]. In the view of these properties, some researchers have hypothesized a key role of vitamin D in reducing the inflammatory microenvironment associated with neoplasia.

Considering our data, 25(OH)D levels were also found as being a potential predictor of the evolution of the disease, as patients who showed a PD had significantly lower 25(OH)D levels. Also, Altieri et al. demonstrated a shorter PFS in patients with 25(OH)D deficiency compared to those with sufficient levels and identified a cut-off of 25(OH)D < 16 ng/mL as associated with higher risk of disease progression [34].

Despite the association observed in our cohort between low 25(OH)D levels and progression, after applying a probabilistic analysis for the PD risk factors, this finding, significant in the univariate analysis, did not maintain statistical significance in the multivariate analysis. On the contrary, the ki67 index and the staging of NEN, as expected, maintained the statistical significance in the multivariate analysis, thus confirming these two factors as the only prognostic elements of PD in our cohort. These data are in line with results from other studies and show that vitamin D could play a role not yet clarified by the retrospective studies available in the literature so far [32,34].

A possible hypothesis of correlation between 25(OH)D levels and value of ki67 could be that at the start of carcinogenesis, 25(OH)D levels could have an impact on biological aggressiveness of the tumor. However, the ki67 value drives the progression of the neoplasm independently from 25(OH)D levels during the natural history of the disease. 

Furthermore, many factors have a greater impact on PFS and OS than 25(OH)D levels; the heterogeneous therapeutic options have a determinant impact both on disease progression and on 25(OH)D levels during follow-up. The retrospective nature of the study does prevent us from drawing conclusions on the real role of vitamin D as a prognostic factor. Vitamin D probably plays a role in the biological aggressiveness of the disease, but it is one of the many variables involved in the prognosis of patients and certainly cannot be considered as a determining factor or used in isolation as an indicator of disease progression. 

Although in our cohort patients who died showed a trend of lower levels of 25(OH)D deficiency, no significant correlation was found regarding the OS, probably due to the low number of events that occurred during our observation period. It is to be taken into consideration that NEN are indolent tumors and the high survival rate of patients could act as a bias in the evaluation of the evolution of the disease over a limited period of time [63].

In our cohort, the evaluation of vitamin D supplementation at diagnosis independent from the 25(OH)D level did not influence PFS and OS. Because of the retrospective nature of the study, it was not possible to obtain data regarding the beginning of the therapy and the treatment compliance before starting of the diagnostic work-up at our clinic. In addition, as previously stated, the indolent nature of this type of neoplasm could delay the diagnosis and the progression, which influences the lack of results.

An interesting study by Robbins et al. about vitamin D supplementation showed a decrease in vitamin D insufficiency in treatment with cholecalciferol 1000–2000 units daily and demonstrated the impact of previous abdominal surgery (not of treatment with somatostatin analogues) on 25(OH)D levels [64]. 

Some other studies evaluated vitamin D supplementation in NEN patients after primary treatments; our study focused on vitamin D supplementation at the time of diagnosis when the patient is still naïve to therapies and surgery that could interfere with absorption [32,64].

The limitations of this study require some consideration. First, its retrospective nature did not allow us to clearly identify 25(OH)D levels as a prognostic factor in NEN. Second, the number of patients included when we stratified our cohort into different subgroups (i.e., according to primary origin) was relatively low. Thirdly, a possible weakness of this study, in common with the whole NEN literature, is the poor number of observed events (death or progression) that affected the survival analyses. This may also be due to the high rate of radically operated patients who achieved a curative resection resulting in a very good prognosis.

Only prospective studies with a long-term follow up, overcoming the biases inherent to retrospective studies, can investigate any causal relationship between 25(OH)D levels and its prognostic role in this disease. Therefore, our results need to be further validated in prospective and larger studies.

The main strengths of this study are represented by a highly selected and well characterized cohort of NENs with precise exclusion criteria to limit possible interferences. In addition, NEN patients and controls were enrolled in the same geographic area to analyze more comparable 25(OH)D levels. Moreover, since our study was mono-institutional, we conducted all the clinical, radiological and biochemical evaluations using the same method with a consequent high reproducibility and increased reliability for comparison.

In conclusion, our study highlights the importance of monitoring 25(OH)D levels in patients with NENs, as its deficiency appeared to be linked to the worst biological aggressiveness of the disease, regardless of tumor origin. Interesting hints on a possible prognostic role of vitamin D in NEN patients needs to be further validated by long-term follow-up prospective studies.

## Figures and Tables

**Figure 1 nutrients-15-03771-f001:**
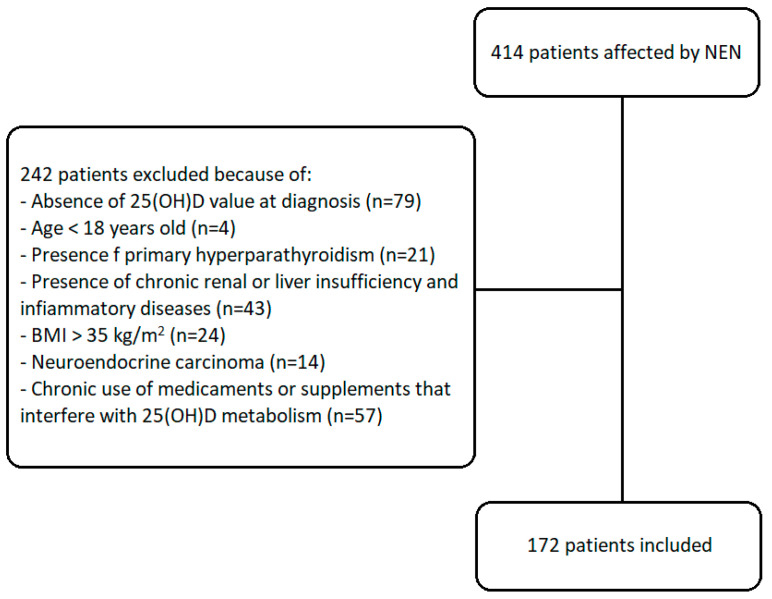
Flow chart of the studied subjects. A total of 172 patients were included from 414 patients with NEN followed up at the Endocrinology Unit of the IRCCS Ospedale Policlinico San Martino, University of Genova.

**Figure 2 nutrients-15-03771-f002:**
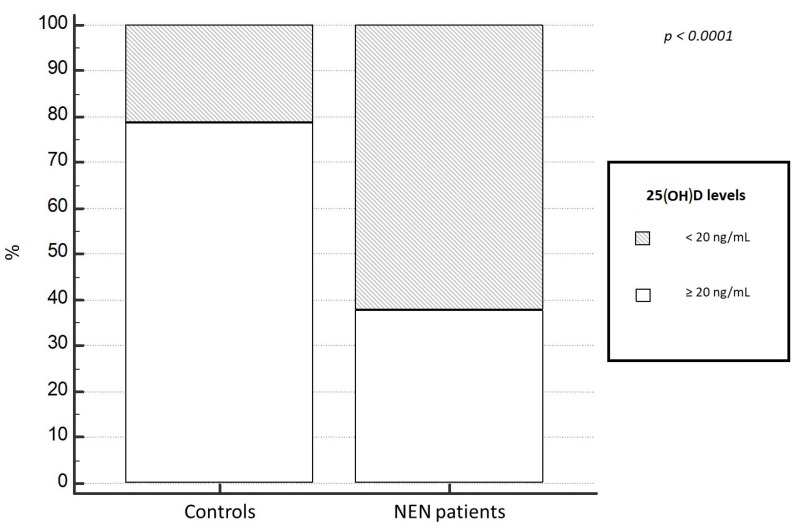
25(OH)D deficiency in NEN patients and controls. NEN, neuroendocrine neoplasms.

**Figure 3 nutrients-15-03771-f003:**
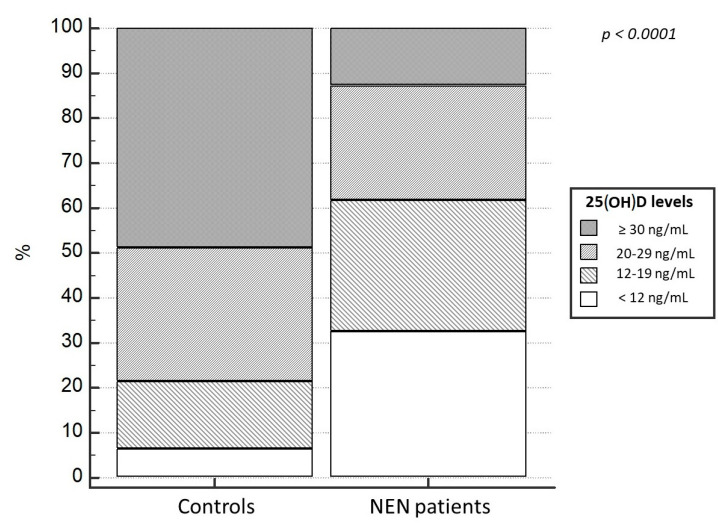
25(OH)D levels in NEN patients and controls. NEN, neuroendocrine neoplasms.

**Figure 4 nutrients-15-03771-f004:**
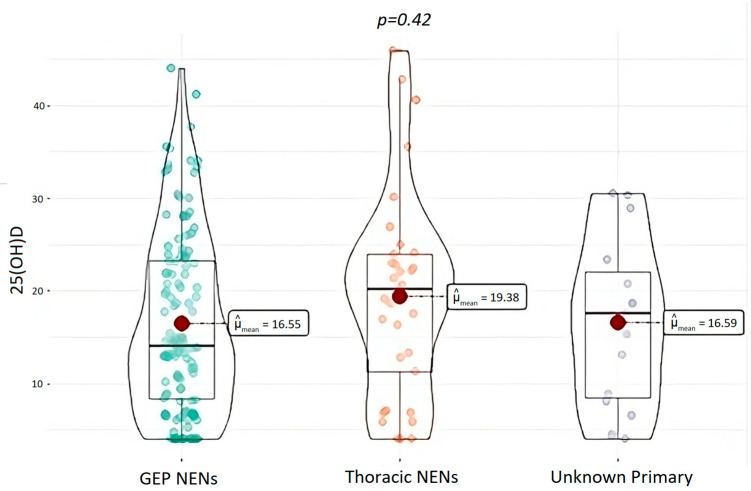
25(OH)D levels in different NEN localizations of primary tumors. NENs, Neuroendocrine Neoplasms; GEP-NENs, Gastroenteropancreatic-Neuroendocrine Neoplasms; and 25(OH)D, 25-OH-vitamin D.

**Figure 5 nutrients-15-03771-f005:**
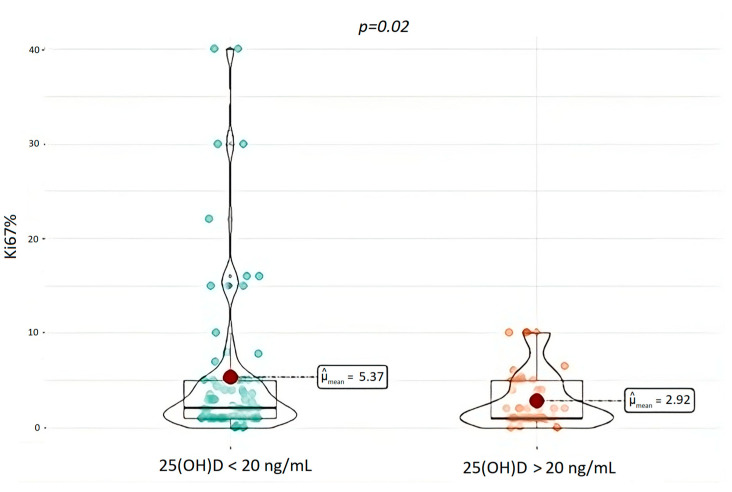
Ki67 index in patients with 25(OH)D < 20 ng/mL and in patients with 25(OH)D > 20 ng/mL. 25(OH)D, 25-OH vitamin D.

**Figure 6 nutrients-15-03771-f006:**
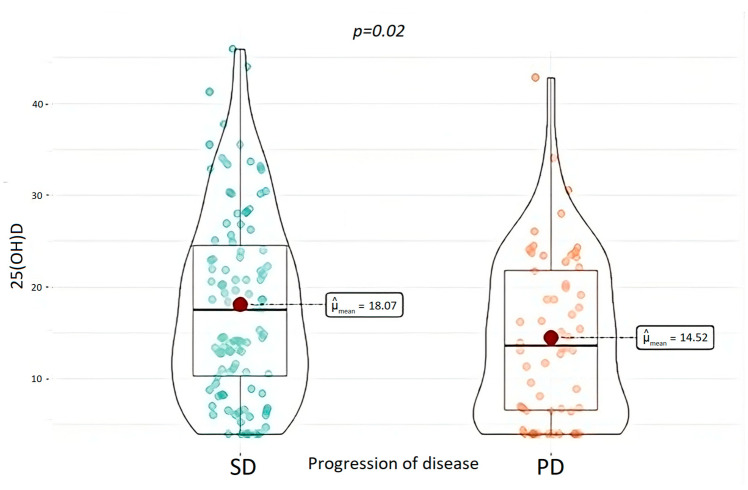
25(OH)D levels in patients with tumor progression and in patients with stable disease. 25(OH)D, 25-OH-vitamin D; PD, tumor progression; and SD, stable disease.

**Table 1 nutrients-15-03771-t001:** Clinical and pathological features of NEN patients (discrete variables). N, number; GEP-NENs, Gastroenteropancreatic-Neuroendocrine Neoplasms; NF, non-functioning; F, functioning; NM, non-metastasis; M, metastasis; NP, non-progression; P, progression; NE, non-exitus; E, exitus; and 25(OH)D, 25-OH-vitamin D.

Patients Features		N		%	
		172		100	
**Sex**					
	Male		73		58
	Female		99		42
**Primary Tumor Localization**					
GEP-NEN	Pancreas		58		33.7
	Ileum		41		23.8
	Stomach		12		7.0
	Appendix		8		4.7
	Colon		5		2.9
	Total		124		72.1
TORACIC-NEN	Lung		30		17.4
	Thymus		3		1.7
	Total		33		19.2
UNKNOWN			15		8.7
**Clinical-Pathological Features**					
FUNCTIONING	NF		113		65.7
	F		50		29.1
SYNDROME	NS		113		65.7
	S		50		29.1
METASTASIS	NM		94		54.7
	M		76		44.2
GRADING	G1		69		40.1
	G2		42		24.4
	G3		2		1.2
	Total		113		
**Clinical Outcome**					
PROGRESSION	NP		106		61.6
	P		53		30.8
EXITUS	NE		154		89.5
	E		18		10.5
**25(OH)D Levels**					
DEFICIENCY	<20		107		62.2
INSUFFICIENCY	20-30		43		25.0
DEFICIT	Total		150		87.2
SUFFICIENCY	>30		22		12.8

**Table 2 nutrients-15-03771-t002:** Clinical and pathological features of NEN patients (continuous variables). MIN, minimum; MAX, maximum; P25, 25° percentile; P75, 75° percentile; SE, standard error; BMI, body mass index; PFS, progression-free survival; OS, overall survival; 25(OH)D, 25-OH-vitamin D; and PTH, parathyroid hormone.

	Mean	Median	Min	Max	P25	P75	SE
**BMI**	24.304	24.300	16.400	34.100	21.000	27.175	0.321
**Age**	59.692	62.500	13.000	89.000	70.000	70.000	1.103
**Ki67%**	5.111	2.000	0.010	90.000	1.000	5.000	0.903
**PFS (months)**	40.545	29.000	1.000	264.000	13.000	52.000	3.189
**OS (months)**	58.351	47.000	1.000	372.000	22.000	77.000	4.110
**Follow-up (months)**	52.628	37.500	1.000	372.000	17.750	73.000	4.021
**25(OH)D**	17.099	15.050	4.000	45.900	8.375	23.425	0.757
**PTH (ng/L)**	39.459	33.000	5.700	347.000	23.000	46.250	2.721
**Calcium (mg/dL)**	9.538	9.500	7.700	11.400	9.200	9.900	0.043
**Phosphorus (mg/dL)**	3.085	3.100	1.600	8.900	2.675	3.400	0.064
**Creatinine (mg/dL)**	0.945	0.900	0.500	8.000	0.700	1.000	0.050
**Size (mm)**	20.485	15.000	1.000	140.000	10.000	25.000	1.684
**Number of lesions**	0.405	0.000	0.000	1.000	0.000	1.000	0.390

**Table 3 nutrients-15-03771-t003:** Statistical analysis showing 25(OH)D levels as a risk factor for disease progression in univariate analysis, not confirmed in multivariate analysis. HR, hazard ratio; IC confidence interval; 25(OH)D, 25-OH-vitamin D; and PTH, parathyroid hormone.

Univariate Analysis				Multivariate Analysis		
	HR	IC95%	*p*	HR	IC95%	*p*
**Male sex**	1.74	0.98–3.1	0.058			Ns
**Multiple lesions**	1.1	0.65–2.26	0.537			
**Ki67**	1.03	1.02–1.05	<0.001	1.034	1.017–1.050	0.006
**Age**	1.02	1.004–1.045	0.022			Ns
**Functioning tumor**	0.97	0.54–1.73	0.919			
**25(OH)D < 20 ng/mL**	1.39	0.49–3.91	0.53			
**25(OH)D levels**	0.96	0.934–0.993	0.016			Ns
**PTH**	1.01	1.00–1.02	0.056			Ns
**Calcium**	1.02	1.002–1.036	0.029			Ns
**IV stadium**	4.23	2.33–7.65	<0.001	4.72	2.04–10.91	<0.001
**25(OH)D supplementation**	0.63	0.36–1.10	0.103			Ns

## Data Availability

The data that support the findings of this study are available from the corresponding author, D.F, upon reasonable request.

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
