# Peer review of "Impact of Vitamin D Deficiency on Tumor Aggressiveness in Neuroendocrine Neoplasms"

_nutrients, 2023, doi:10.3390/nu15173771_

Round 1

Reviewer 1 Report

The treatment options for neuroendocrine neoplasms include surgery, chemotherapy, radionuclide therapy, somatostatin analogue therapy, molecular targeted therapies, alpha-interferon therapy, and inhibitors of serotonin production. Treatment methods and lower patient exposure are being sought. Currently, nutritional and vitamin status is a neglected area in patients with NENs, so the topic is interesting.

however, it seems necessary to introduce a few suggestions and draw the reader's attention to other issues in order to improve the reception of the article

The following issues should be addressed at the outset:

1. In the randomized clinical trials (RCT) of neuroendocrine tumors, 22 different therapy strategies were compared, stating that there are a number of effective therapies with different safety profiles available to patients doi:10.1001/jamaoncol.2018.6720

2. As a result of the illness, the consumption of some nutrients can be reduced, leading to nutritional deficiencies and resulting in malnutrition. Vitamin D deficiency was observed in 100% of men and women. Moreover, both men and women experienced the deficiency of vitamin E, folates and niacin. https://doi.org/10.3390/nu12071961

The NET patient group includes people with metabolic disorders, which often lead to the development of diabetes and obesity. These diseases are also associated with vitamin D deficiency

3. bioactive compounds, including antioxidants, vitamins, flavonoids are an important element in the fight against NEN, increasing the success of therapy 

4. vitamin D deficiency is common in populations living in temperate and cold climates, which should be considered in the introduction, because regardless of the intake and needs of the body, it affects the level of this vitamin

METHODS

a very large study group is a great advantage of this study despite the rejection of more than half of the patients in the group

5. Please provide the reasons why most of the patients were rejected from the analysis

6. Characteristics of the studied patients (initial) with neuroendocrine tumors would be required (age, sex, origin, type of NENs, anthropometric measurements) in order to demonstrate that the narrowed group was not subject to deliberate selection

7. was the vitamin D measurement done only once? was the average of several measurements taken at intervals during treatment ?

Results

8. this section lacks information on supplementing patients with vitamin D

9. What % were patients, what dose was most commonly used, and how this affected the overall vitamin D levels in these patients. In which group were the supplementing patients

Well-conducted discussion and indicated limitations of the study and differences in interpretation. 

10. It is also worth referring to the literature on supplementation https://doi.org/10.1080/01635581.2018.1470650

best regards

Author Response

We thank the Reviewer for the suggestion.
Please see the attachment.

We also edit the bibliografy and highlight the modification.

Reviewer 2 Report

General considerations

This manuscript is well-written on a relevant and current subject, with few publications. The English language level is quite adequate. However, some elucidations are necessary for a better understanding of the content of the text.

Specific considerations

Title

As the title indicates, what results did the authors obtain that relate vitamin D deficiency with tumor aggressiveness in neuroendocrine neoplasms? (Line 1).

Abstract

"Our study highlights the importance of monitoring 25(OH)D levels in patients with NENs, as its deficiency appeared to be linked to worst biological aggressiveness and worst outcome" (Line 37). What results do the authors obtain that allow this conclusion? Do the authors consider "disease progression" and "tumor aggressiveness" equivalent?

Introduction

"In the last years, some studies have evaluated vitamin D levels in NENs and specifically in gastroenteropancreatic NENs (GEP-NENs), most of them demonstrating a higher prevalence of vitamin D deficit in those patients (29–32)" (Line 71). Were these results significant? If so, it is necessary to highlight these findings as "significant."

Subjects and Methods

The authors initially studied a population of 414 patients and analyzed a population of 172 patients (Line 89). It is necessary to display the reasons for reducing the initial group of patients through a flowchart.

As the authors indicate, this study is retrospective (Line 89). How was the determination of the plasmatic level of vitamin D obtained? Was this determination included in the examination protocol for patients with NENs? Were blood samples stored? When exactly were the samples taken?

The eventual supplementary use of vitamin D by patients and controls before measuring the levels of this vitamin used in the study was not considered an exclusion criterion. Could this not be considered a bias? (Line 108)

What was the reason for excluding patients with poor differentiation at histologic examination (NEC) (Line 121)?

"The statistical analysis revealed significant differences between groups (p < 0.05) using both parametric and non-parametric tests" (Line 157). Hypothetically, this statement is a result and should be moved to the Results section unless the authors refer to biodemographic parameters between NENs and control groups.

RESULTS

How did the authors calculate the number of subjects (50) to compose the control group? (Line 162).

"Considering the 25(OH)D cut-off values of 12 and 20 ng/ml at the time of the diagnosis, biological aggressiveness was therefore evaluated" (Line 227). Please, specify what these values indicate.

"Patients with 25(OH)D <20 ng/ml had significantly lower Ki67 (p=0.02) compared to the ones with 25(OH)D levels

above 20 ng/ml, as shown in fig. 4." (Line 231). This result seems contradictory to the other results and the authors' conclusions.

As the authors explain in conclusion, in patients with NENs, the deficiency of 25(OH)D levels appeared to be linked to the disease's worst biological aggressiveness and worst evolution (Line 392). So, how does this conclusion align with the results obtained in this study that indicated "no significant correlation was found among 25(OH)D deficit and the number of metastases, the staging or the grading of the disease" (Line 240) and "mean 25(OH)D levels did not differ between patients with G2 tumors compared to those with G1 (17.0 ± 9.5 vs 16.1 ± 10.2 ng/mL)" (Line 242)? Additionally, no significant difference was found depending on 25(OH)D levels considering the low number of events, as far as the decease (Line 258) and the tumor progression and "exitus" were not significantly influenced by vitamin D supplementation (Line 260). Furthermore, neither the presence of 25(OH)D deficiency (p=0.52) nor its supplementation (p=0.09) resulted in influencing the PFS in the present cohort, and even the overall survival (OS) did not result significantly influenced by 25(OH)D levels or supplementations (Line 261).

The authors showed that in the univariate analysis, low 25(OH)D levels were found to be a risk factor for disease progression, together with the Ki67 index and the stage of the disease at the diagnosis. However, the 25(OH)D levels did not confirm their role as a predictor of progression in the multivariate analysis, whereas Ki67 and 267 disease staging were confirmed as significant predictors in the multivariate analysis. This result indicates that the vitamin D level depends on other independent variables, which restricts its isolated use as an indicator of disease progression. Authors need to consider these findings from their study and contextualize them in the Discussion section and the manuscript's conclusions.

Table 3: Tables need to be self-explanatory. The legend must be completed.

The quality of the English language is adequate.

Author Response

(The authors gave the same response as above.)

Reviewer 3 Report

In this manuscript, the authors analyzed the prevalence of vitamin D in the pathogenesis and outcome of neuroendocrine neoplasms (NENs) in a cohort of 172 patients with NENs and 50 healthy volunteers with a negative history of neoplasia or other inflammatory diseases. In conclusion, they find that vitamin D deficiency is highly prevalent among NENs and is associated with higher ki67 and disease progression, Moreover, the authors also stated that vitamin D deficiency appeared to be linked to worst biological aggressiveness and worst outcome in patients with NENs. However, they data showed that the PFS and OS were not significantly influenced by vitamin D. Therefore, it should be explained why their conclusion is different with this results. Besides, the current manuscript is not well prepared. Some errors should be revised before the manuscript become to be publishable. The following are some comments for the manuscript in detail.

1. The clinical features of 50 healthy volunteers should be presented. Only state that “matched for age, sex, BMI, and geographical provenience with the patients of NEN group” is not enough.

2. There are many methods for multiple group comparisons after the ANOVA with parametric data with normal distribution. The authors should state it clear which method was used.

3. In line 145, the “PFS” should be “PFS time” according to their definition.

4. The title of the figures should under the figures.

5. The authors use the word “tumour” in the discussion part, but the “tumor” in the other parts. It should be consistent.

Author Response

(The authors gave the same response as above.)
